# *Rickettsia* Infection Benefits Its Whitefly Hosts by Manipulating Their Nutrition and Defense

**DOI:** 10.3390/insects13121161

**Published:** 2022-12-15

**Authors:** Ze-Yun Fan, Yuan Liu, Zi-Qi He, Qin Wen, Xin-Yi Chen, Muhammad Musa Khan, Mohamed Osman, Nasser Said Mandour, Bao-Li Qiu

**Affiliations:** 1Chongqing Key Laboratory of Vector Insects, College of Life Sciences, Chongqing Normal University, Chongqing 401331, China; 2Guangdong Laboratory for Lingnan Modern Agriculture, Guangzhou 510642, China; 3Engineering Research Center of Biocontrol, Ministry of Education Guangdong Province, South China Agricultural University, Guangzhou 510640, China; 4Department of Plant Protection, Faculty of Agriculture, Suez Canal University, Ismailia 41522, Egypt

**Keywords:** *Bemisia tabaci*, endosymbiont, entomopathogenic fungus, parasitoid, insecticide

## Abstract

**Simple Summary:**

*Rickettsia* is a maternally transmitted endosymbiotic bacterium that infects most insect species. In the current study, we investigated the biological and physiological effects of *Rickettsia* infection on whitefly, *Bemisia tabaci*. Our results revealed that infection with *Rickettsia* increased the fertility, survivorship, and shortened the nymphal developmental duration of whitefly *Bemisia tabaci*. *Rickettsia* infected *B*. *tabaci* had significantly higher glycogen, soluble sugar and trehalose contents than those of *Rickettsia* negative *B. tabaci* individuals. When exposed to the entomopathogenic fungus *Akanthomyces attenuatus* and the insecticides imidacloprid and spirotetramat, *Rickettsia* infested *B. tabaci* had lower mortality rates and higher semi-lethal concentrations (LC_50_). The parasitism by *Encarsia formosa* was also reduced by *Rickettsia* infection.

**Abstract:**

Endosymbionts play an essential role in the biology, physiology and immunity of insects. Many insects, including the whitefly *Bemisia tabaci*, are infected with the facultative endosymbiont *Rickettsia*. However, the mutualism between *Rickettsia* and its whitefly host remains unclear. This study investigated the biological and physiological benefits of *Rickettsia* infection to *B*. *tabaci*. Results revealed that infection of *Rickettsia* increased the fertility, the survival rate from nymph to adult and the number of female whiteflies. In addition, this facilitation caused a significant reduction in nymphal developmental duration but did not affect percentage rate of egg hatching. *Rickettsia* infected *B*. *tabaci* had significantly higher glycogen, soluble sugar and trehalose contents than *Rickettsia* negative *B*. *tabaci* individuals. *Rickettsia* also improved the immunity of its whitefly hosts. *Rickettsia* infested *B*. *tabaci* had lower mortality rates and higher semi-lethal concentrations (LC_50_) when exposed to the fungus *Akanthomyces attenuatus* and the insecticides imidacloprid and spirotetramat. The percentage of parasitism by *Encarsia formosa* was also reduced by *Rickettsia* infection. Overall, *Rickettsia* infection benefits *B*. *tabaci* by improving the nutritional composition of its host, and also protects *B*. *tabaci* by enhancing its resistance towards insecticides (imidacloprid and spirotetramat), entomopathogenic fungi (*A*. *attenuatus*) and its main parasitoid (*E. formosa*); all of which could significantly impact on current management strategies.

## 1. Introduction

Endosymbiotic bacteria are prevalent within invertebrates. Some of them supply diet limited nutrients to their insect hosts [1,2], some can alter their host’s tolerance to extreme environmental stress [3,4], while others are reproduction manipulators [5]. Insects form a substantial part of many ecological networks and often close associations with maternally inherited, intracellular bacteria. Insect facultative symbionts are often reproduction manipulators and the diversity by which intracellular bacteria manipulate their host reproduction is considerable and ranges from parthenogenesis, male-killing, cytoplasmic incompatibility to the functional feminization [6,7,8]. It is important to explore both the advantages and disadvantages of a symbiotic relationship between insects and symbionts, since some relationships could aid in the management of insect pests, while others may facilitate the host and significantly reduce the effectiveness of a given pest management strategy.

Whitefly *Bemisia tabaci* is a destructive pest containing more than 40 cryptic species including two invasive populations, the Middle East-Asia Minor 1 (MEAM1, formerly B biotype) and the Mediterranean (MED, formerly Q biotype) [9,10]. Apart from direct feeding, it causes serious crop losses worldwide by indirectly transmitting viruses [11]. In China, *B*. *tabaci* MEAM1 and MED are well distributed across 31 Provinces and have caused significant economic losses since the mid-1990s and early 2000s, respectively. Whitefly is the major pest of many glasshouse vegetables, ornamentals and field crops in China [12,13]. Various characteristics of the whitefly including rapid reproduction, production of wax powder and rapid development of chemical resistance have led to great challenges in their control and management [14].

Regarding the endosymbionts of *B*. *tabaci*, in addition to the primary symbiont *Portiera*, several genera of facultative symbionts, such as *Arsenophonus*, *Cardinium*, *Hamiltonella*, *Hemipteriphilus*, *Rickettsia* and *Wolbachia* have been recorded within various cryptic species of the *B*. *tabaci* complex [15,16,17,18,19,20]. Previous studies have shown that, *Rickettsia* is abundant in many insects including whitefly pests in nature with stable and high frequencies of infection [21,22,23,24]. It can manipulate host reproduction by causing male-killing in some ladybird beetles or parthenogenesis in eulophid wasps [25,26,27]. In *B*. *tabaci*, it can modify the adaptability of its whitefly hosts to their environment, such as enhancement in stress resistance to temperature [28], and altering the survival rate, fecundity and offspring of the whiteflies [29]. However, the effects of *Rickettsia* infection may vary between different *B*. *tabaci* cryptic species, or even within the same cryptic species but from different geographical populations. Some studies have revealed the impact of infection of *Rickettsia* on nutrition changes of *B*. *tabaci* [30]. However, very few reports come from a multicomponent systems interaction viewpoint; taking *Rickettsia* associated defense against fungi, parasitoids and insecticides into one study [31,32,33,34]. This lack of information limits our knowledge concerning the *Rickettsia*-whitefly interaction and the development of subsequent management strategies.

This study firstly investigated the impact of *Rickettsia* on the biology of its whitefly host *B*. *tabaci* MEAM1, characterizing how the *Rickettsia* infection affects the biology of its host by altering its nutrition. Secondly, this study assessed the contribution of *Rickettsia* infection to the immunity of *B*. *tabaci* against entomopathogenic fungi, insecticides and parasitism. The study aimed to reveal the specific undescribed interactions of *Rickettsia* and *B*. *tabaci* and give further insight and understanding in the continual development of management strategies against *B*. *tabaci*.

## 2. Materials and Methods

### 2.1. Plants

Cotton plants (*Gossypium hirsutum* L. var. Lumianyan no. 32) were used in the current study to rear *B*. *tabaci* populations. Cotton seeds were sown in plastic pots (12 cm diameter × 15 cm height) containing a soil–sand mixture (10% sand, 5% clay and 85% peat). Plants were reared in a glasshouse (26 ± 1 °C, 16:8 h L:D photoperiod) in a pest- and pesticide-free environment and watered as required. Plants were used for experiments at their 6–8 expanded leaf stage.

### 2.2. Insects

*Bemisia tabaci* MEAM1 cryptic species, were initially collected from eggplant (*Solanum melongena*) grown at the training farm of South China Agricultural University (SCAU) in Guangzhou, China in 2016. Populations were then reared on cotton plants under standard laboratory conditions at 26 ± 1 °C, 60% relative humidity (RH) and a photoperiod of 16:8 h (L:D). Mitochondrial CO1 gene sequencing was used to check and maintain the purity of the *B*. *tabaci* populations [35]. Two *B*. *tabaci* MEAM1 populations, *Rickettsia* positive (*R^+^* MEAM1) and *Rickettsia* negative (*R^−^* MEAM1) were set up and maintained according to the methods of Liu et al. [36]. In brief, population screening was conducted for single-pair purification using cotton as the host plant. Random whiteflies, newly emerged and not yet mated, were selected in order to identify their sex (female and male) under a stereomicroscope (Zeiss SteREO Discovery, Zeiss, Oberkochen, Germany). Following this, one pair of whiteflies were released into a leaf cage which was attached onto a clean cotton leaf to allow egg laying for 6 days. After this, the parent adult whiteflies were recaptured from the cage and examined for the presence of *Rickettsia* using the *Rickettsia*-specific primers (*16S rRNA*, *gltA* and *Pgt*) [18,37]. The above steps were repeated to purify the population of *B*. *tabaci* MEAM1 until the *B*. *tabaci* MEAM1 glasshouse population was numerous enough for experimental trials. To ensure the purity of each population, approximately 100 adult whiteflies were selected and checked for the presence/absence of *Rickettsia* respectively every month.

The parasitoid *Encarsia formosa* was first collected from parasitized *B*. *tabaci* nymphs on tomato plants at Beijing Academy of Agriculture and Forestry Sciences in 2008. Subsequent offspring were reared on cotton plants infested by *Rickettsia* positive *B*. *tabaci* in a separate glasshouse.

### 2.3. Entomopathogenic Fungus and Insecticides

*Akanthomyces attenuatus* (previously *Lecanicillium attenuatus*) SCAUDCL53 strain (NCBI accession No. MH558279) was provided by the Engineering Research Center of Biocontrol, Ministry of Education in SCAU, which was initially collected from Fujian Province in 2016. The *A*. *attenuatus* was inoculated on a Potato Dextrose Agar (PDA) medium in 9 cm diameter Petri dishes and sealed with parafilm. Plates were incubated for eight days under the conditions outlined by Khan et al. [38]. Following sporulation, the spores were scraped into a sterilized dry conical bottle containing 50 mL 0.05% Tween-80 and thoroughly shaken via a magnetic stirrer. The conidial suspension was filtered and poured into a new, sterilized dry conical flask. A hemocytometer (Neubauer) was used to determine the concentration of the stock solution which was diluted into five concentrations ranging from 1 × 10^8^, 1 × 10^7^, 1 × 10^6^, 1 × 10^5^ and 1 × 10^4^ conidia/mL.

The two insecticides, imidacloprid (95% WP, Anhui Huaxing Chemical Co., Ltd., Maanshan, China) and spirotetramat (97% WP, Hebei Weiyuan Biochemical Co., Ltd., Shijiazhuang, China) were used to assess the effect of *Rickettsia* infection on the chemical resistance of *B*. *tabaci* MEAM1.

### 2.4. Effect of Rickettsia on Development and Reproduction of Bemisia tabaci

Healthy and fully expanded cotton leaves were marked on the cotton plants and covered with leaf cages (diameter: 7 cm; height: 7 cm). Five pairs of *Rickettsia* positive (*R^+^*) and *Rickettsia* negative (*R^−^*) *B*. *tabaci* adults (2–4 days old) were separately released into a leaf cage to allow egg-laying for five days. After five days, the number of eggs produced by the mating pairs were counted under a stereomicroscope (Zeiss SteREO Discovery). This experiment investigating fecundity was repeated in nine parallel replications for both *R^+^* and *R^−^* whiteflies.

In addition, 20–30 pairs of *R^+^* and *R^−^ B*. *tabaci* adults (2–4 days old) were separately released into a leaf cage to allow egg-laying for 24 h. After 24 h the adult whiteflies were removed. The development and survival of the *B*. *tabaci* nymphs were then observed every 24 h. Following the emergence of adult whiteflies, sex identification was made under a stereomicroscope (Zeiss SteREO Discovery, Zeiss, Oberkochen, Germany). The treatments were repeated in three parallel replications.

### 2.5. Effect of Rickettsia on the Nutritional Changes of Bemisia Tabaci

The nutritional contents of *B*. *tabaci* were determined by using 100 pairs of *R^+^* and *R^−^ B*. *tabaci* adults respectively. A total of ten cotton leaves were caged to introduce whiteflies of each *Rickettsia* status for oviposition for 24 h. After the subsequent emergence of *R^+^* and *R^−^* whitefly adults (2 h old), specimens were collected in an Eppendorf tube, weighed 10 mg per sample and frozen in liquid nitrogen. Frozen samples were then ground and homogenized in PBS (pH 7.4). This crude homogenate was centrifuged at 10,000 rpm at 4 °C for 10 min and stored at −80 °C for further experimentation.

The protein contents of *R^+^* and *R^−^ B*. *tabaci* adults were measured according to the protocol provided by the manufacturer Beyotime Biotechnology. The absorbance was read in a Microplate Spectrophotometer (XMark™, BIO-RAD, Hercules, CA, USA). Bicinchoninic acid (BCA) was used to determine Cu^+^ at a wavelength of 562 nm [39].

The anthrone method was used to estimate the soluble sugar and glycogen content as outlined by Halhoul and Kleinberg [40] and trehalose as outlined by Ferreira et al. [41] at 630 nm wavelength by using a microplate analyzer. The same supernatant was used for all the experiments.

### 2.6. Effect of Rickettsia on the Defense of Bemisia tabaci against Akanthomyces attenuatus

A total of ten healthy cotton plants were taken, and three fully expanded cotton leaves were selected from each plant and covered with individual leaf cages. There were forty pairs of *R^+^* and *R^−^ B*. *tabaci* adults that were separately released into a leaf cage of different plants, respectively. After 48 h of egg-laying, the whiteflies were removed. Emerged nymphs were reared up to the 4th instar on the respective leaf. A total of one hundred nymphs were randomly selected per instar, with excess nymphs removed from the leaves via a fine camel hairbrush. Leaves with whitefly nymphs were plucked from the plants and immersed in the conidia suspension with concentrations mentioned above for 15s and then allowed to air-dry at room temperature as outlined in Cuthbertson et al. [42]. Leaves dipped in 0.05%Tween-80 were used as controls. Following this, the leaves were placed in Petri dishes containing 1% water agar medium, covered with a thin plastic layer with small puncture holes for aeration. All the treatment and control experiments were repeated in three parallel replications. All the Petri dishes were placed in separate climate chambers (PQX-250, Jintan Experimental Instrument Co. Ltd., Jiangsu, China) to avoid contamination at identical temperature (26.0 ± 1 °C), relative humidity (70–85%) and photoperiod (14:10 (L:D)); the light intensity was maintained at approximately 3000 Lux. Survival data were collected daily over the following seven days.

### 2.7. Effect of Rickettsia on the Defense of Bemisia Tabaci against the Parasitoid

As outlined above, three healthy and expanded cotton leaves were selected from one cotton plant and covered with leaf cages. Fifty pairs of *R^+^* and *R^−^ B*. *tabaci* adults were separately released into the leaf cages. Following egg-laying for 24 h, the whiteflies were removed. Approximately 160 nymphs were randomly selected per instar, with excess nymphs removed using a fine camel hairbrush. Eight females of *E*. *formosa* (5 days old) were introduced into the leaf cage for 24 h before being removed. All the treatments were repeated five times. The parasitism rate, developmental duration, and emergence rate (%) of *E*. *formosa* were subsequently recorded.

### 2.8. Effect of Rickettsia on the Resistance of Bemisia tabaci to Insecticides

For the toxicity assay, seven geometrically progressive concentrations ranging from 3.125 mg/L, 6.25 mg/L, 12.5 mg/L, 25 mg/L, 50 mg/L, 100 mg/L and 200 mg/L of imidacloprid and spirotetramat were diluted in water. The dip impregnation method was used to determine the toxicity of imidacloprid and spirotetramat to second instar nymphs of *R^+^* and *R^−^ B*. *tabaci*. When the nymphs developed to second instar, 100 nymphs on one leaf were randomly selected and plucked from the plant, immersed entirely in the different pesticide concentrations of imidacloprid and spirotetramat for 10s, then dried at room temperature, again following method of Cuthbertson et al. [42]. For controls, individual leaves were dipped in ddH_2_O. Treated leaves were again placed in Petri dishes as outlined above. The data for survival of *R^+^* and *R^−^ B*. *tabaci* nymphs were collected daily over the following five days.

The residual method was used to determine the toxicity of imidacloprid and spirotetramat to *R^+^* and *R^−^ B*. *tabaci* adults. Plants and *B*. *tabaci* adults were obtained as outlined above. In a 20 mL tube, 0.5 mL of each concentration of imidacloprid and spirotetramat were added separately; the tubes were then physically shaken well to apply the pesticide to the wall of the tube evenly. Following 1 min of shaking, the remaining pesticide was discarded, and the tubes air-dried. Again, tubes washed in ddH_2_O were used as controls. A total of fifteen pairs of *R^+^* and *R^−^ B*. *tabaci* adults were then placed into each tube, respectively, for 30 min before being released into the new leaf cages. All the treatment and control experiments were repeated in three parallel replications. Whitefly mortality was recorded after six hours.

### 2.9. Statistical Analyses

Statistical analyses were performed by using SAS software (v.8.01). Data were tested for normality (Shapiro–Wilks test) and homogeneity of variance (Levene’s test) before using parametric tests. Egg hatch ability, the mortality rate of whitefly, the parasitism rate and the emergence rate of parasitoids among different treatments were arcsine transformed wherever the data did not conform to a normal distribution. The biological data, nutrition contents and parasitism were compared among treatments using a *t*-test. The cumulative corrected mortality rate (%) of whiteflies caused by the entomopathogenic fungus and insecticides respectively were compared among treatments using two-way ANOVA. Tukey’s post-hoc test assessed the mean difference between and among the treatments at *p* < 0.05. Significant differences between treatments were estimated at *p* < 0.05, *p* < 0.01, and *p* < 0.001 significance levels. Graphical work was done via GraphPad Prism 5 (GraphPad, La Jolla, CA, USA).

## 3. Results

### 3.1. Effect of Rickettsia on Development and Reproduction of Bemisia tabaci

Our results showed that both the fecundity and egg hatching rate of *R^+^ B*. *tabaci* are higher than that of *R^−^ B*. *tabaci*, with the difference in fecundity being significant (t_16_ = 12.31, *p* = 0.0001, t_4_ = 0.79, *p* = 0.47; Figure 1a,b). The developmental period of *R^+^ B*. *tabaci* F1 generation was also significantly shorter than that of *R^−^ B*. *tabaci* individuals (t_4_ = 2.88, *p* = 0.045; Figure 1c), but their survivorship from egg to adult was higher than the *R*^-^ individuals (t_4_ = 3.12, *p* = 0.03; Figure 1d). In addition, the percentage of females in the *R^+^* F1 generation was significantly higher than that of *R^−^* F1 generation (t_4_ = 4.794, *p* = 0.0087; Figure 1e), and the average longevity of *R^+^* F1 female adults was significantly longer than that of *R^−^* F1 female adults (t_4_ = 4.585, *p* = 0.01; Figure 1f). Therefore, we can conclude that *Rickettsia* plays a positive role in terms of the fecundity, the number of females and survival rate in the *B*. *tabaci* MEAM1 population.

### 3.2. Effect of Rickettsia on the Nutritional Components of Bemisia tabaci

The presence of *Rickettsia* had clear effects on the nutritional components of *B*. *tabaci*. The contents of glycogen (t_4_ = 2.89, *p* = 0.04), soluble sugar (t_4_ = 4.10, *p* = 0.015) and trehalose (t_4_ = 3.48, *p* = 0.025) were all significantly elevated in the *R^+^ B*. *tabaci* compared to that of *R^−^* individuals (Figure 2a–c). However, there was no significant change between the protein concentrations of *R^+^* and *R^−^ B*. *tabaci* populations (t_4_ = 0.05, *p* = 0.96) (Figure 2d).

### 3.3. Effect of Rickettsia Persistence on Bemisia tabaci Defense against Akanthomyces attenuatus

The bioassay results showed that the *A*. *attenuatus* SCAUDCL53 isolate has high pathogenicity to all instar nymphs of *R^+^* and *R^−^ B*. *tabaci* (Figure 3). At five days after infection, compared with healthy 3rd instar nymphs (Figure 3a-1), the fungus-infected 3rd instar nymphs (Figure 3a-2) were wrapped in white mycelium and had a change in body color. Comparing healthy 2d old pupae (Figure 3b-1), the fungus-infected 2d old pupae (Figure 3b-2) were again wrapped in white mycelium. Here, the body became dried out and again a color change was evident. When comparing healthy newly emerged adults (Figure 3c-1), the fungus-infected newly emerged adults (Figure 3c-2) showed symptoms such as being wrapped in white mycelium, changes in body color, unresponsiveness and in several cases the body became dried up. The mortality rate increased with the increase in conidial suspension concentration; highest mortality rate was at a concentration of 1 × 10^8^ conidia/mL. Overall, the mortality rate of *R^+^ B*. *tabaci* was distinctly lower than that of *R^−^ B*. *tabaci* (Figure 4). No matter the age of the whitefly treated, the conidial suspension concentration and *Rickettsia* all significantly affected the mortality of the whitefly (Appendix A).

Bioassay results revealed, when infecting the MEAM1 nymphs, a higher semi-lethal concentration (LC_50_) is required for *R^+^ B*. *tabaci* than *R*^−^ *B*. *tabaci* to get 50% mortality. This indicates that the *Rickettsia* negative MEAM1 nymphs were more susceptible to *A*. *attenuatus* infection (Table 1).

### 3.4. Effect of Rickettsia Infection on Parasitism Rate of Encarsia formosa

*Rickettsia* infection distinctly increased the defense ability of *B*. *tabaci* against parastization from the endoparasitoid *E*. *formosa*. The average parasitism rate of *E*. *formosa* in *R^+^ B*. *tabaci* reduced approximately 26% compared to those in *R^−^ B*. *tabaci* (t_8_ = 2.50, *p* = 0.037; Figure 5a). Also, the generational developmental duration of *E*. *formosa* progeny in *R^+^ B*. *tabaci* nymphs was about 6.56% shorter than those in *R^−^ B*. *tabaci* nymphs (t_8_ = 3.38, *p* = 0.0097; Figure 5b). However, there was no significant effect on the emergence rate of *E*. *formosa* progeny that developed in the *R^+^* and *R^−^ B*. *tabaci* nymphs (t_8_ = 0.88, *p* = 0.40; Figure 5c).

### 3.5. Effect of Rickettsia Infection on Insecticide Resistance of Bemisia tabaci

With an increase in imidacloprid concentrations, *B*. *tabaci* second instar nymphs and adults’ mortality significantly increased. Also, the mortality of *R^+^ B*. *tabaci* second instar nymphs and adults were both lower than those of *R^−^ B*. *tabaci* second instar nymphs (Figure 6a) and adults (Figure 6b). The semi-lethal concentrations (LC_50_) of imidacloprid to *R^+^ B*. *tabaci* second instar nymphs and adults were higher than those of *R^−^ B*. *tabaci* second instar nymphs and adults (Table 2). In addition, the bioassay results showed that when second instar whitefly nymphs were treated with imidacloprid, the concentration of imidacloprid significantly affected the mortality of the whitefly; but there was no association with the infection status of *Rickettsia*. However, when the whitefly adults were treated, both the concentration of imidacloprid and *Rickettsia* infection status significantly affected the mortality of the whitefly (Appendix A).

Different results were observed when spirotetramat was used against *R^+^* and *R^−^ B*. *tabaci* second instar nymphs (Figure 6c) and adults (Figure 6d). The semi-lethal concentrations (LC_50_) of spirotetramat to *R^+^ B*. *tabaci* second instar nymphs and adults were higher than those of *R^−^ B*. *tabaci* second instar nymphs and adults (Table 2). When second instar whitefly nymphs were treated with spirotetramat, both the concentration of spirotetramat and *Rickettsia* infection status significantly affected the mortality of the whitefly; significant interactions were observed. However, when whitefly adults were treated the concentration of spirotetramat significantly affected the mortality of the whitefly, but there was no association with the *Rickettsia* infection status (Appendix A). In general, all the bioassay data indicated that *Rickettsia* infection enhanced the whitefly host’s resistance to insecticides.

## 4. Discussion

Insect bacterial endosymbionts affect insect hosts’ biological, physiological and ecological traits, including their adaptation to temperature stress, immunity and resistance ability against entomopathogenic fungi and natural enemies; endosymbionts may also affect the development, survival and reproductive pattern of their insect hosts [28,43,44,45]. Although *Rickettsia* species have been verified in their function as primary nutritional symbionts and reproductive manipulators [37,46], their role in the vast majority of hosts is unknown. The *Rickettsia* in *B*. *tabaci* MEAM1 is in the well-defined *bellii* clade [36,47], which has also shown to positively influence various fitness measures of *B*. *tabaci*, including the induction of a higher reproduction rate and a female-biased sex ratio [22,48]. In this study, we demonstrated the role of *Rickettsia* focusing on its resistance ability against entomopathogenic fungi, a natural enemy and several chemical pesticides.

It has been confirmed that *Rickettsia* can impact whiteflies in multiple ways including reproduction, development and survivorship [49]. For example, infected *Rickettsia* whiteflies produce more offspring that survive to adulthood at greater rates and develop more quickly compared with uninfected whiteflies [22]. Similarly, Chiel et al. [29] and Shi et al. [45] reported that infection of *Rickettsia* significantly shortened the developmental period of their *B. tabaci* hosts. Our results revealed that *R^+^ B. tabaci* have higher fecundity, survival rate, number of females and higher longevity than the *R^-^ B. tabaci*, while there were shorter developmental periods for *R^+^ B. tabaci*. All our results in the current study further confirmed the above findings.

Endosymbionts are also known as male-killers or to be female-biased (convert non-transmitting male hosts into transmitting females through feminization of genetic males and parthenogenesis induction) [50], among which, *Rickettsia* has been revealed to manipulate host reproduction, either by killing male offspring as embryos (male-killing) or by inducing parthenogenesis [26]. Our results also showed that *R^+^ B*. *tabaci* produced more female offspring than the *R^−^* population. It has been reported that, *Rickettsia* can raise the fitness of infected female indirectly by manipulating host reproduction, either by killing male offspring as embryos (male-killing) or by inducing parthenogenesis. The result of nucleotide sequencing of the 16S rRNA gene in Hagimori et al. [51] indicated that the parasitoid *Neochrysocharis formosa* (Westwood) is infected with a *Rickettsia* bacterium, which appears to be causative of the thelytokous parthenogenesis (in which mothers produce only female offspring from unfertilized eggs). This is the first finding of parthenogenesis-induction by *Rickettsia* among insects. Moreover, *Neochrysocharis formosa* (Westwood) (Hymenoptera: Eulophidae) males infected with *Rickettsia* produced by antibiotic treatment exhibited the same courtship behaviors as the arrhenotokous males, but at a lower rate, and did not produce fertilized progeny [52].

Endosymbionts have been reported to affect the nutritional contents of insect hosts [53]. For instance, the breakdown of glycogen generates glucose, which enters the glycolytic pathway being converted into pyruvate. This process leads to ATP generation and provides energy for insect activities [54]. It has been reported that the function of soluble sugar is to offer energy for the hosts’ muscles when an insect is walking or escaping [55,56]. Thus, alteration in soluble sugar content may cause a significant change in the normal functioning of the organism. Trehalose is an important disaccharide in all biological forms and provides energy for growth, metamorphosis, stress recovery, chitin synthesis, and insect flight [57]. From the elevated level of glycogen, soluble sugar and trehalose, we hypothesized that *Rickettsia* infestation also affects the physiology of *B*. *tabaci*. Endosymbionts can promote insect fitness by contributing to nutrition [58], they play a prominent role in insect nutritional ecology by aiding in digestion of food or providing nutrients that are limited or lacking in the diet [59,60]. Lv et al. [61] reported that *Buchnera aphidicola* helps the pea aphid *Acyrthosiphon pisum* to overcome the nutritional deficiency of a plant-based diet. In our study, due to increased egg-laying, *B*. *tabaci* produces more energy reserves to support reproduction and other daily metabolic activities.

Endosymbionts also play an important role in the enhancement or detraction of the defense system of the pest host against different management strategies, for example, insecticides [62], entomopathogenic fungi [63] and biological control parasitoids [42,64]. Our results showed that *Rickettsia* enhanced resistance of *B*. *tabaci* to imidacloprid, spirotetramat, it has been found that insecticide resistance was increased in hosts infected with some symbionts, but we speculate that the enhancement or decrease of insecticide resistance may depend on the endosymbiont-insecticide association, for example, *Rickettsia* increased the resistance of whiteflies to acetamiprid and spiromesifen, but not diafenthiuron [65]; *Rickettsia* coexisting with another symbiont, *Arsenophonus*, was shown to confer insecticide resistance to acetamiprid in *B*. *tabaci*, but did not affect susceptibility to diafenthiuron [62]. In addition, Pan et al. [66] reported that the thiamethoxam-susceptible population of *B*. *tabaci* harbored more *Portiera* and *Hamiltonella* than the thiamethoxam-resistant population, whereas the thiamethoxam-resistant population of *B*. *tabaci* harbored more *Rickettsia* than the thiamethoxam-susceptible population.

We reached a consensus currently that non-chemical control measures became the alternative and best choice for the management of insect pests, due to increasing resistance to chemical pesticides [67]. Some studies indicated several facultative endosymbionts of the pea aphid have been implicated in increasing their host resistance to pathogenic fungi [43,68,69,70]. Panteleev et al. [63] revealed that females of *Drosophila melanogaster* infected with *Wolbachia* were more resistant to the fungus *Beauveria bassiana* (an insect pathogen) than uninfected females; infected females also exhibited changes in oviposition substrate preference. Hendry et al. [32] also reported that *Rickettsia* infected *B*. *tabaci* exhibited a decreased mortality rate due to the entomopathogenic bacteria *Pseudomonas syringae* compared to *Rickettsia* negative *B*. *tabaci*. Our results revealed that *R*^+^ *B*. *tabaci* individuals showed a significant mortality reduction when *A*. *attenuates* was applied, and to each specific stage of *B*. *tabaci*, a higher concentration was necessary to manage *R*^+^ individuals compared to the *R^−^* individuals. Endosymbionts also protect their host against parasitoids. The endosymbiont *Buchnera aphidicola* protects *Acyrthosiphon pisum* against the hymenopteran parasitoid *Aphidius ervi* by causing high mortality in developing parasitoid larvae [43]. *Regiella insecticola* was also reported to protect its aphid hosts against the parasitoids *Aphidius colmani* and *Aphidius asychis* [71,72]. *Hamiltonella* did not reduce the susceptibility of aphid to two species of parasitoids (*A. ervi* and *Ephedrus plagiator*) and did not affect the fitness of wasps that successfully completed development, but it may reduce the risk of parasitism in its aphid hosts by making them less attractive to searching parasitoids [73]. All these studies along with our findings indicate that, although differing in symbiont species, endosymbionts may share the same functional contribution to their insect hosts.

In conclusion, results from this study and those of previous studies suggest that *Rickettsia* infestation benefits *B. tabaci* by aiding in enhanced reproduction, higher survival and faster development by improving its host’s nutritional composition. *Rickettsia* infection improved its host’s fitness by enhancing its resistance towards insecticides (imidacloprid and spirotetramat), entomopathogenic fungus (*A. attenuatus*) and parasitoid (*E. formosa*). This study is useful in understanding the role of endosymbionts within an insect host. Endosymbionts can affect the fitness of their host and they play important roles in protecting their host from environmental stress, such as natural enemies and toxins. However, there are still several unanswered questions that need to be addressed: (1) bacterial endosymbionts are common in insects, so potentially symbiont-mediated protection exists in many insect species, thus, how common is this phenomenon in nature and is this effect the same in different insects? (2) Endosymbionts can enhance their resistance towards insecticides, entomopathogenic fungi and parasitoids, but the breadth of mechanisms that underlie how the symbionts provide protection is still largely unknown. (3) How should we adjust subsequent pest control strategies in response to these characteristics of symbiotic bacteria? All these aspects should be further investigated in the future, to support development of novel strategies of pest biological control.

## Figures and Tables

**Figure 1 insects-13-01161-f001:**
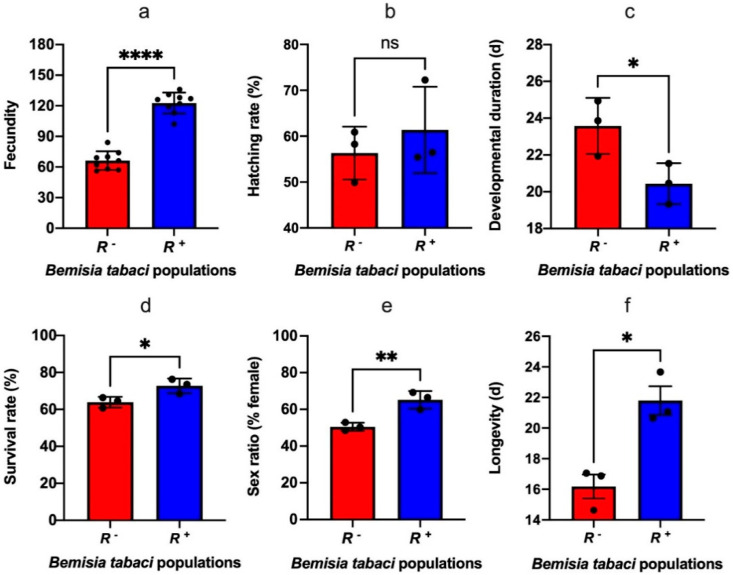
The effect of *Rickettsia* infection on the fecundity (**a**), hatching rate (**b**), developmental du-ration (**c**), survival rate from egg to adult (**d**), sex ratio (% female) (**e**) and longevity (**f**) of *Bemisia tabaci* MEAM1 cryptic species. *R^+^*: *Rickettsia* positive population; *R^−^*: *Rickettsia* negative population. Data were compared among treatments using *t*-test, and stars over the bars *, **, **** signify differences were significantly different at 0.05, 0.01 and 0.0001 levels respectively, ns signifies differences were not significant.

**Figure 2 insects-13-01161-f002:**
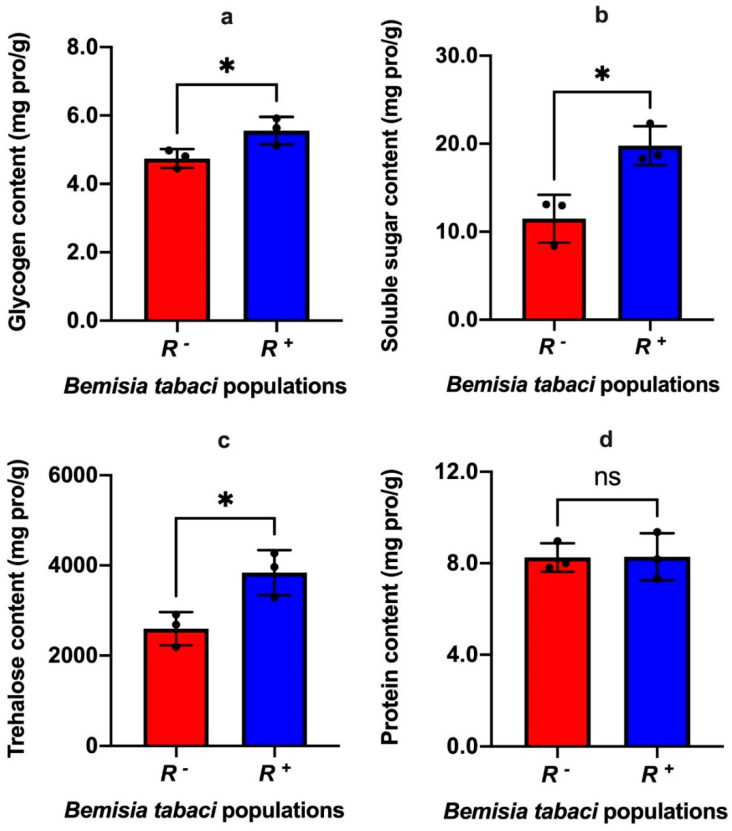
The effect of *Rickettsia* infection on the glycogen content (**a**), soluble sugar content (**b**), trehalose content (**c**) and protein content (**d**) of *Bemisia tabaci* MEAM1 cryptic species. *R*^+^: *Rickettsia* positive population, *R^−^*: *Rickettsia* negative population. Data were compared among treatments using *t*-test, and stars over the bars * signify differences were significantly different at 0.05 level respectively, ns signifies differences were not significant.

**Figure 3 insects-13-01161-f003:**
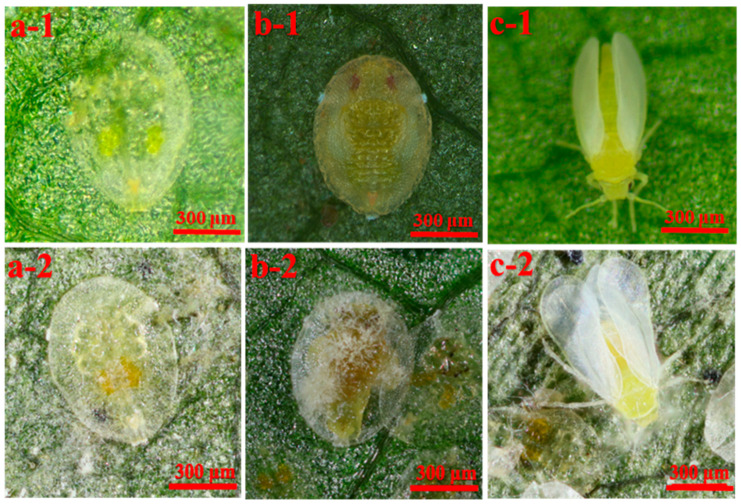
The infection phenotype of *Bemisia tabaci* MEAM1 nymphs treated with *Akanthomyces at-tenuatus* (1 × 10^8^ conidia/mL). Panels (**a-1**,**a-2**) were healthy 3rd nymphs and the fungus-infected 3rd nymphs, (**b-1**,**b-2**) were healthy 2d age pupae and the fungus-infected 2d age pupae, (**c-1**,**c-2**) were healthy newly emerged adults and the fungus-infected newly emerged adults of *Rickettsia* negative *B*. *tabaci* on the 5th day after infection.

**Figure 4 insects-13-01161-f004:**
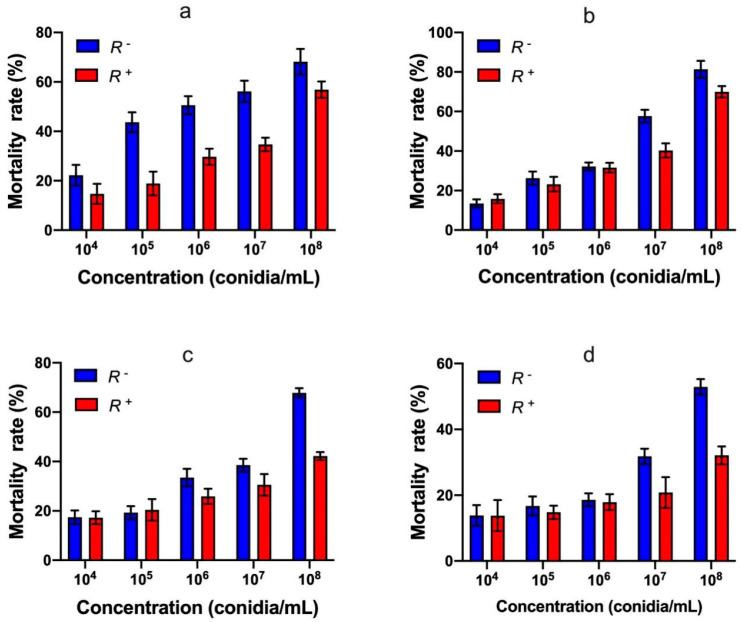
The effect of *Akanthomyces attenuatus* SCAUDCL53 on the mortality rate of the first instar (**a**), second instar (**b**), third instar (**c**) and fourth instar (**d**) nymphs of *Rickettsia* positive (*R*^+^) and *Rickettsia* negative (*R^−^*) *B*. *tabaci* MEAM1 cryptic species.

**Figure 5 insects-13-01161-f005:**
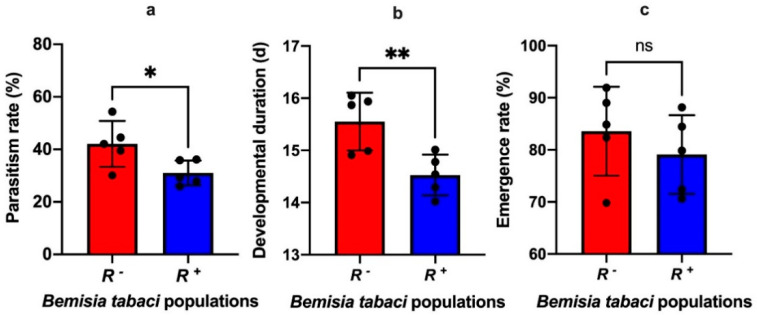
The effects of *Rickettsia* infection in *B*. *tabaci* MEAM1 on the parasitism of *Encarsia formosa*. (**a**) parasitism rate, (**b**) developmental duration of *E*. *formosa* F1 larvae, (**c**) emergence rate of *E*. *for-mosa* F1 adults. *R*^+^: *Rickettsia* positive population, *R*^−^: *Rickettsia* negative population. Data were com-pared among treatments using *t*-test, and stars over the bars *, ** indicate that differences were significantly different at 0.05 and 0.01 levels respectively, ns indicate that differences were not significant.

**Figure 6 insects-13-01161-f006:**
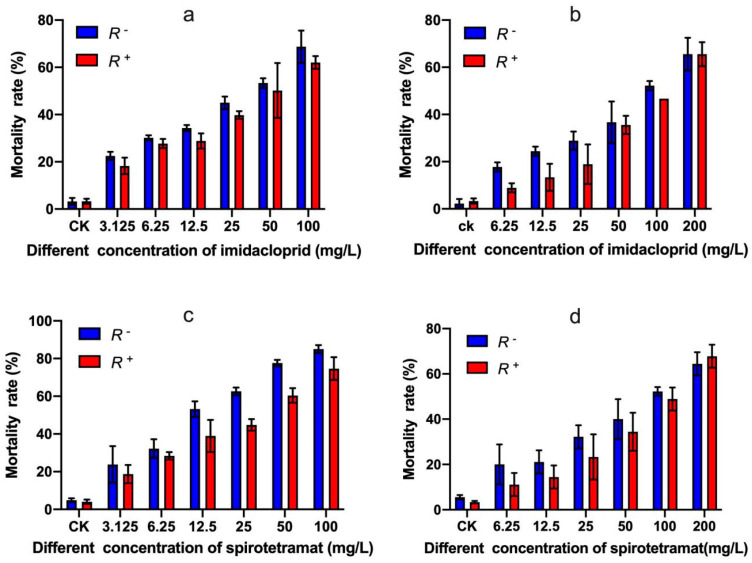
The effects of imidacloprid and spirotetramat on the mortality of second instar nymphs (**a**,**c**), adults (**b**,**d**) of *Rickettsia* positive and *Rickettsia* negative *B*. *tabaci* MEAM1 cryptic species. *R*+: *Rickettsia* positive population, *R*^−^: *Rickettsia* negative population. Control treatment (CK) was ddH_2_O.

**Table 1 insects-13-01161-t001:** Virulence of *Akanthomyces attenuatum* SCAUDCL53 against different *Rickettsia* positive and *Rickettsia* negative developmental stages of *B*. *tabaci*.

Instar	*Rickettsia* +/−	LC_50_ (95% CI Conidia/mL)	Regression Virulence Model	χ^2^	*p*
1st	*R* * ^−^ *	1.52 × 10^6^ (5.77 × 10^5^ − 4.30 × 10^6^)	Y = 0.273x − 1.687	4.18	0.24
	*R^+^*	6.18 × 10^7^ (1.93 × 10^7^ − 3.87 × 10^8^)	Y = 0.297x − 2.317	2.64	0.45
2nd	*R* * ^−^ *	3.19 × 10^6^ (1.80 × 10^6^ − 6.0 × 10^6^)	Y = 0.488x − 3.172	4.80	0.19
	*R^+^*	1.28 × 10^7^ (5.62 × 10^6^ − 3.80 × 10^7^)	Y = 0.357x − 2.54	6.19	0.10
3rd	*R* * ^−^ *	1.56 × 10^7^ (2.37 × 10^6^ − 1.29 × 10^9^)	Y = 0.350x − 2.52	6.60	0.09
	*R^+^*	2.23 × 10^9^ (1.65 × 10^8^ − 1.36 × 10^12^)	Y = 0.188x − 1.754	0.50	0.92
4th	*R* * ^−^ *	1.82 × 10^8^ (4.65 × 10^7^ − 1.79 × 10^9^)	Y = 0.294x − 2.426	5.42	0.14
	*R^+^*	4.93 × 10^11^ (3.41 × 10^9^ − 8.07 × 10^19^)	Y = 0.151x − 1.768	1.404	0.70

**Table 2 insects-13-01161-t002:** Toxicity of imidacloprid and spirotetramat against different developmental stages of *Rickettsia* positive and *Rickettsia* negative *B*. *tabaci*.

Pesticide	Instar	*Rickettsia +*/*−*	LC_50_ (95% CI) mg/L	Regression Virulence Model	χ^2^	*p*
Imidacloprid	Adult	*R* * ^−^ *	88.28 (65.14 − 132.22)	Y = 0.87x − 1.70	2.22	0.70
		*R^+^*	106.32 (83.28 − 144.95)	Y = 1.20x − 2.43	1.82	0.77
	2nd nymph	*R* * ^−^ *	34.89 (25.20 − 53.33)	Y = 0.76x − 1.17	3.38	0.50
		*R^+^*	44.28 (31.79 − 69.72)	Y = 0.79x − 1.30	2.05	0.73
Spirotetramat	Adult	*R* * ^−^ *	97.97 (75.97 − 132.55)	Y = 1.15x − 2.28	2.03	0.73
		*R^+^*	120.14 (57.00 − 1336)	Y = 0.71x − 1.47	0.83	0.66
	2nd nymph	*R* * ^−^ *	13.24 (10.68 − 16.19)	Y = 1.24x − 1.39	2.69	0.61
		*R^+^*	24.83 (19.46 − 32.56)	Y = 1.00x − 1.39	0.55	0.97

## Data Availability

Not applicable.

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
