# Peer review of "Rickettsia Infection Benefits Its Whitefly Hosts by Manipulating Their Nutrition and Defense"

_insects, 2022, doi:10.3390/insects13121161_

Round 1

Reviewer 1 Report

In this article, Fan et al described the impact of Rickettsia on the biology of its host whitefly, showing how the Rickettsia infection affects the biology of its host by altering its nutrition, and regulating the immunity of whitefly to fungi, insecticides and parasitism. This is an interesting, thorough, and well-written paper. There are some minor errors should be corrected before acceptance for publication.

Abstract

Line 27 : Change mutual benefit mechanism to “mutualism”

Line 28-29: Change the sentence as “ this study investigates the biological------------------ to B. tabaci”

Line 32 Change “B. tabaci have ” to “B. tabaci had”

Line 35: add the full annotation of LC50

Line 39: the last sentence of abstract needs a bit more illustartion.

Introduction:

Line 53: Change reproductive to “reproduction”

Line 54 : Delete “a”

Line 62” Change “other relationships” to “others”

Line 62 Change “The whitefly” to “Whitefly”

Line 92-98: Authors should try to improve the need of experiment a bit further.

Materials and methods:

Line 104-105: What do you mean by ambient temperature and photoperiod.

Line 108-109: Change the sentence as “ Bemisia tabaci MEAM1 cryptic species initially collected from eggplant (Solanum melongena) grown at the training farm of South China Agricultural University (SCAU) in Guangzhou, China in 2016”.

Line 128: Replace the term “Sporangium powder” with proper words.

Line 131-132: add information of equipment used during conidial count .

Line 141: Add dimensions/information of leaf cage.

Line 149: stereomicroscope brand??

Line 166: Change fungus to “insect pathogenic fungus”

Line 171: Change “A hundred “ to One hundred”

Line 220-221: Whayt was the need to use tukeys’ post hoc test as only two treatments were compared. Also add reference for graphpad prism 5.

Results:

Line 231: Female ratio should be written as “ male:female ratio”

Line 254: Add full annotation for LC50.

Line 289-290. Please rewrite the sentence.

Discussion:

Authors should try to explain a little more about the possible mechanisms involved in reported changes in different whitefly parameters.

Concluding paragraph should add a little more information about the limitations as well as future implications of the study.

Please re-check as if all the refrence are as per journal format.

Author Response

Dear Sir/Madam,

We fully appreciate all the constructive comments and suggestion received from you and the respected reviewers. Please now find enclosed a revised version of the manuscript. We have revised the manuscript carefully according all of the comments and suggestions. We believe that our revision work has greatly added to the quality of the manuscript.

The Reviewer comments are now listed below with our response given to each one. I trust this revised manuscript now proves satisfactory. All co-authors are in agreement with the revised version.

Yours sincerely,

Some of comments, raised by the respected reviewer 1, have been already revised before we submitted current version to insects. And the new comments and our revision are listed below:

Point 1: Line 27 : Change mutual benefit mechanism to “mutualism”

Have done.    

Point 2: Line 28-29: Change the sentence as “this study investigates the biological------------------ to B. tabaci

Have done.

Point 3: Line 32 Change “B. tabaci have” to “B. tabaci had”

Have done.

Point 4: Line 35: add the full annotation of LC50

Have done.

Point 5: Line 39: the last sentence of abstract needs a bit more illustration.

Have done.

 Point 6: Line 53: Change reproductive to “reproduction”

Have done.

Point 7: Line 54 : Delete “a”

Have done.

Point 8: Line 62” Change “other relationships” to “others”

Have done.

Point 9: Line 62 Change “The whitefly” to “Whitefly”

Have done.

Point 10: Line 92-98: Authors should try to improve the need of experiment a bit further.

Have done, we improved the need of experiment.

Point 11: Line 104-105: What do you mean by ambient temperature and photoperiod.

Have corrected, temperature is 26±1℃ and photoperiod is 16:8 h L:D.

Point 12: Line 108-109: Change the sentence as “Bemisia tabaci MEAM1 cryptic species initially collected from eggplant (Solanum melongena) grown at the training farm of South China Agricultural University (SCAU) in Guangzhou, China in 2016”.

Have done. we revised this sentence from line 88 to 90.

Point 13: Line 128: Replace the term “Sporangium powder” with proper words.

Have done, we have revised the words before submission to insects.

Point 14: Line 131-132: add information of equipment used during conidial count.

Have done, we have revised the words before submission to insects.

Point 15: Line 141: Add dimensions/information of leaf cage.

Have done, we added information of leaf cage in line 128.

Point 16: Line 149: stereomicroscope brand??

We added information of stereomicroscope brand in line 131.

Point 17: Line 166: Change fungus to “insect pathogenic fungus”

We revised “fungus” to “Akanthomyces attenuates” in line 156.

Point 18: Line 171: Change “A hundred “to One hundred”

Have done

Point 19: Line 220-221: What was the need to use tukeys’ post hoc test as only two treatments were compared. Also add reference for graphpad prism 5.

The biological data, nutrition contents and parasitism were compared among treatments using a t-test. The cumulative corrected mortality rate (%) of whiteflies caused by the entomopathogenic fungus and insecticides respectively were compared among treatments using two-way ANOVA and Tukey's post-hoc test assessed the mean difference between and among the treatments at P<0.05. When only two treatments were compared don’t need to use tukeys’ post-hoc test. We added details about graphpad prism in line 218.

Point 20: Line 231: Female ratio should be written as “male:female ratio”

We measured the percentage of females, not the male:female ratio.

Point 21: Line 254: add full annotation for LC50.

Have done. We added full information for LC50 in the whole paper.

Point 22: Line 289-290. Please rewrite the sentence.

Have done. We have revised this sentence before we submit manuscript to insects.

Point 23: Authors should try to explain a little more about the possible mechanisms involved in reported changes in different whitefly parameters. Concluding paragraph should add a little more information about the limitations as well as future implications of the study.

Have done. We have revised these comments before we submit manuscript to insects.

Point 24: Please re-check as if all the reference are as per journal format.

Have done.

Reviewer 2 Report

The study “Rickettsia infection benefits its whitefly hosts by manipulating their nutrition and defence” has significant findings for the scientific literature. There is a question about whether the symbiotic bacteria are the same in different species of whiteflies, and whether the symbiotic bacteria have the same function. Before the acceptance of this manuscript, I have some comments on the current version.

1.     Abstract: LC50 change to LC50

2.     Line 53 change “resistance towards chemicals has” to “chemical resistance” change have

3.     Line 94, according to “the methods” of Liu et al. (2022)

4.     Line 96 change “which were newly emerged and not yet mated, were selected in order” to “newly emerged and not yet mated, were selected”

5.     Line 108, which cryptic species of B. tabaci was used for parasitoid rearing?

6.     Line 110, is Akanthomyces attenuatus a natural pathogen of whitefly?

7.     Line 121-122, why the authors select imidacloprid and spirotetramat these two kinds of pesticides?

8.      Line 153, add space between 630 nm

9.     Line 155, I suggest: against Akanthomyces attenuatus instead of entomopathogenic fungus

10. Line 172 change “Survival data was” to “Survival data were”

11. Line 232-233, R+ and R- should be R+, R-

12. Line 248, again, against Akanthomyces attenuatus

13. Line 255 change “a change in color” to “a color change

14.  Line 293 change “the mortality of B. tabaci 2nd instar nymphs and adults both” to ” B. tabaci 2nd instar nymphs and adults mortality

15.  Line 325 change “many biological, physiological and ecological traits of insect host” to “insect hosts’ biological, physiological and ecological traits”

16.  Line 418 “through improving the nutritional composition of its host” to “by improving its host’s nutritional composition”

17. The figure 4 is insert into the legend, please adjust

18.  Table 1 Rickettsia, R+, R- all should be italic

19.  Again, Figure 6 and Figure 7 insert into their legends, have the authors checked the PDF before confirming the submission?

20.  Table 2 Rickettsia, R+, R- all should be italic

21.  The words in Table A1 A2 such as: insecticide, age, source should be capital for the first letter.

22. Please re-check as if all the references are as per journal format.

Author Response

Dear Sir/Madam,

We fully appreciate all the constructive comments and suggestion received from you and the respected reviewers. Please now find enclosed a revised version of the manuscript. We have revised the manuscript carefully according all of the comments and suggestions. We believe that our revision work has greatly added to the quality of the manuscript.

The Reviewer comments are now listed below with our response given to each one. I trust this revised manuscript now proves satisfactory. All co-authors are in agreement with the revised version.

Yours sincerely,

Point 1: Abstract: LC50 change to LC50

Have done.

Point 2: Line 53 change “resistance towards chemicals has” to “chemical resistance” change have

Have done.

Point 3: Line 94, according to “the methods” of Liu et al. (2022)

Have done.

Point 4:   Line 96 change “which were newly emerged and not yet mated, were selected in order” to “newly emerged and not yet mated, were selected”

Have done.

Point 5:  Line 108, which cryptic species of B. tabaci was used for parasitoid rearing?

We used Rickettsia positive B. tabaci to rear parasitoids. We add some details in line 108.

Point 6: Line 110, is Akanthomyces attenuatus a natural pathogen of whitefly?

Yes. Akanthomyces attenuates is a natural pathogen of whitefly. It affects several biological indicators of whitefly, such as growth, development and detoxification enzymes.

Point 7:  Line 121-122, why the authors select imidacloprid and spirotetramat these two kinds of pesticides?

Imidacloprid and spirotetramat are high efficiency and broad spectrum and these are important insecticides for chemical control of Bemisia tabaci.

Point 8:   Line 153, add space between 630 nm

Have done.

Point 9:  Line 155, I suggest: against Akanthomyces attenuatus instead of entomopathogenic fungus

Have done.

Point 10:   Line 172 change “Survival data was” to “Survival data were”

Have done.

Point 11:   Line 232-233, R+ and R- should be R+, R-

Have done.

Point 12:   Line 248, again, against Akanthomyces attenuates

Have done.

Point 13:   Line 255 change “a change in color” to “a color change

Have done.

Point 14:   Line 293 change “the mortality of B. tabaci 2nd instar nymphs and adults both” to” B. tabaci 2nd instar nymphs and adults mortality”

Have done.

Point 15:   Line 325 change “many biological, physiological and ecological traits of insect host” to “insect hosts’ biological, physiological and ecological traits”

Have done.

Point 16:  Line 418 “through improving the nutritional composition of its host” to “by improving its host’s nutritional composition”

Have done.

Point 17:  The figure 4 is insert into the legend, please adjust

Have done.

Point 18:  Table 1 Rickettsia, R+, R- all should be italic

Have done.

Point 19:  Again, Figure 6 and Figure 7 insert into their legends, have the authors checked the PDF before confirming the submission?

Have done.

Point 20:  Table 2 Rickettsia, R+, R- all should be italic

Have done.

Point 21:  The words in Table A1 A2 such as: insecticide, age, source should be capital for the first letter.

Have done.

Point 22:   Please re-check as if all the references are as per journal format.

Have done.

Reviewer 3 Report

The manuscript by Fan et al describes the phenotypic variation and biological differences of a Rickettsia-free breeding line of the whitefly Bemisia tabaci. The aim is to elucidate the variable ecological roles that this endosymbiont can have in various insects. The work is based on a previous manuscript by Liu et al. [Reference 36], which has established the symbiont-free insect line beforehand. While this manuscript focusses on the biological differences regarding insect fitness.

The authors investigated various factors and demonstrated that Rickettsia infection can influence reproduction and fertility, developmental time and survival rate as well as the resistance to pathogens and pesticides, hence various positive effects on insect fitness for this pest insect. Overall this manuscript is well written and results are presented coherently. The authors conclude that these effects are probably caused du to the nutritional influences on whiteflies which were also measurable in terms of carbohydrate content.

Minor comments:

Several figures have been placed within the figure legends and need to be corrected e.g. Fig 4, Fig 6 and Fig 7.

Fig.2 b) c) f) Axis starts not at zero.

Figure numbering should be revised. There is no reference to the graphical abstract, while the first main figure of the manuscript is called “Figure 2”.

Author Response

Response to Reviewer 3’ comments

Dear Sir/Madam,

We fully appreciate all the constructive comments and suggestion received from you and the respected reviewers. Please now find enclosed a revised version of the manuscript. We have revised the manuscript carefully according all of the comments and suggestions. We believe that our revision work has greatly added to the quality of the manuscript.

The Reviewer comments are now listed below with our response given to each one. I trust this revised manuscript now proves satisfactory. All co-authors are in agreement with the revised version.

Yours sincerely,

The manuscript by Fan et al describes the phenotypic variation and biological differences of a Rickettsia-free breeding line of the whitefly Bemisia tabaci. The aim is to elucidate the variable ecological roles that this endosymbiont can have in various insects. The work is based on a previous manuscript by Liu et al. [Reference 36], which has established the symbiont-free insect line beforehand. While this manuscript focusses on the biological differences regarding insect fitness.

The authors investigated various factors and demonstrated that Rickettsia infection can influence reproduction and fertility, developmental time and survival rate as well as the resistance to pathogens and pesticides, hence various positive effects on insect fitness for this pest insect. Overall this manuscript is well written and results are presented coherently. The authors conclude that these effects are probably caused due to the nutritional influences on whiteflies which were also measurable in terms of carbohydrate content.

Minor comments:

Point 1: Several figures have been placed within the figure legends and need to be corrected e.g. Fig 4, Fig 6 and Fig 7.

Have done.

Point 2: Fig.2 b) c) f) Axis starts not at zero.

If axis stats at zero, figures look out of scale and unsightly, if needed, we will change.

Point 3: Figure numbering should be revised. There is no reference to the graphical abstract, while the first main figure of the manuscript is called “Figure 2”.

Have done, we revised the figure numbering.
